# State-of-Health Estimate for the Lithium-Ion Battery Using Chi-Square and ELM-LSTM

Jianfeng Jiang [1], Shaishai Zhao [2] and Chaolong Zhang [2,*]

1   School of Intelligent Engineering Technology, Jiangsu Vocational Institute of Commerce, Nanjing 211168, China; kaxing002@163.com
2   School of Electronic Engineering and Intelligent Manufacturing, Anqing Normal University, Anqing 246011, China; zhaoshaishai@126.com
*   Correspondence: zhangchaolong@126.com

**Abstract:** The state-of-health (SOH) estimation is of extreme importance for the performance maximization and upgrading of lithium-ion battery. This paper is concerned with neural-network-enabled battery SOH indication and estimation. The insight that motivates this work is that the chi-square of battery voltages of each constant current-constant voltage phrase and mean temperature could reflect the battery capacity loss effectively. An ensemble algorithm composed of extreme learning machine (ELM) and long short-term memory (LSTM) neural network is utilized to capture the underlying correspondence between the SOH, mean temperature and chi-square of battery voltages. NASA battery data and battery pack data are used to demonstrate the estimation procedures and performance of the proposed approach. The results show that the proposed approach can estimate the battery SOH accurately. Meanwhile, comparative experiments are designed to compare the proposed approach with the separate used method, and the proposed approach shows better estimation performance in the comparisons.

**Keywords:** lithium-ion battery; health monitoring; chi-squared statistic; extreme learning machine; long short-term memory neural network



## 1. Introduction

Lithium-ion batteries play an imperative role in electrification fields including electric vehicles, electric grids, and portable electronic devices, thanks to their excellent properties in weight, power density, cycle life and self-discharging rate [1–6]. However, despite considerable progress in battery chemistry and material, the degeneracy phenomenon is always keeping in line with the usage and aging of lithium-ion batteries [7,8]. Therefore, it is essential to accurately monitor the real-time battery health status avoiding catastrophic hazards and improving battery durability and safety. As one of the main state parameters of the battery, state-of-health (SOH) is generally applied to quantify the degree of battery aging and health status. However, the complexity and non-linearity of the electrochemical mechanism have always been a technical difficulty in estimating the battery SOH accurately [9,10]. Meanwhile, it constitutes the major incentive for taking advantage of various advanced theories. A variety of methods were proposed to obtain accurate estimates of the battery SOH, which can group into three categories: direct measurement, adaptive approaches and data-driven methods.

Coulomb counting (CC), open circuit voltage (OCV) and impedance spectroscopy are commonly used to directly estimate the battery SOH. CC means that full charging and discharging processes are performed to acquire the battery static SOH [11,12]. Extensive tests are conducted to obtain a relationship between SOH and OCV in the OCV approach [13,14]; impedance spectroscopy uses a wide frequency spectrum to determine SOH [15]. This scheme is simple and straightforward, but it is time consuming and only

for specialized laboratory environment, which led to the limited application of direct measurement methods.

The adaptive approaches, based on various advanced algorithms, calibrate the battery SOH by identifying the model parameters. The model is generally an electrochemical model or equivalent circuit model. Due to the high complexity and incapable of directly solving the issue of battery SOH estimation, the electrochemical model is still not popular in practical application [16,17]. The extended Kalman filter (EKF) [18] and particle filter (PF) [19] are generally utilized for the SOH estimation based on the equivalent circuit model. Nonetheless, their effectiveness and adaptability are sensitive to the credibility and robustness of the prescribed battery model, which is affected seriously by the degree of the complexity of electronic systems and uncertain environments of practical application.

Data-driven methods attract more attention since they can describe the complex battery degradation process without the need for in-depth mechanism research. These methods automate the battery SOH estimation work through mapping external characteristics to capacity loss with available historic data of batteries. This type of methods mainly includes four principal classes: (1) direct mapping from the aging cycle to SOH; (2) mapping from achievable variables to SOH; (3) signal processing and (4) statistical metrics [20]. The method of directly mapping from the aging cycle to SOH has good nonlinear mapping [21]. Artificial neural network [22], fuzzy logic [23], and support vector machine (SVM) [24] are commonly utilized to map from achievable variables to SOH through a black-box model, which is automatically build up with the input data to reflect the relationship between the battery SOH and the measured data. Signal processing includes incremental capacity analysis (ICA) [25], differential voltage analysis (DVA) [26], and differential thermal voltammetry (DTV) [27]. This kind of differential signal requires to be further manipulated using regression technique. Statistical metrics involve statistical dependency analysis, the sample entropy, and so forth. However, these approaches have disadvantages of poor generation and sensitive to the quantity and quality of battery data. As a deep-learning neural network, the long short-term memory (LSTM) neural network has been utilized to estimate the battery SOH, because it is able to store and update the information efficiently for a long period of time without vanishing gradient [28]. Extreme learning machine (ELM) is a novel single-hidden layer feedforward neural network learning method, and it has been used for the battery SOH and state-of-charge (SOC) estimation over the advantages of high precision and high self-adaptability [29].

To improve the precision of SOH estimation results, classical methods usually wield a huge amount of data to construct the battery model. However, the measured battery data are often subject to disturbances, measurement errors, stochastic load and other unknown factors in the actual application works. Responses to the challenges, this paper is concerned with ensemble learning-enabled battery SOH indication and estimation. The insight that motivates this work is that the chi-square of battery voltages of each constant current-constant voltage phrase and mean temperature could reflect the battery capacity loss effectively. The ensemble algorithm included by ELM and LSTM neural network (ELM-LSTM) is further utilized to capture the underlying correspondence between the SOH, mean temperature, and chi-square of battery voltages. NASA battery data and battery pack data are used to demonstrate the estimation procedure and performance of the proposed approach. The results show that the proposed approach can estimate the battery SOH accurately. Meanwhile, comparative experiments are designed to compare the proposed approach with the separate used method, and the proposed approach shows better estimation performance in the comparisons.

The structure of this paper is arranged as follows: Sections 2 and 3 describe the definition of SOH and chi-square, respectively. The ELM-LSTM algorithm is presented in Section 4. Section 5 discusses the experiment procedure and results. The key conclusions are finally summarized in Section 6.

## 2. SOH Definition

SOH of the battery can be defined in different ways such as internal resistance and capacity. In this paper, the SOH is estimated using the capacity approach and defined as:

$$\text{SOH} = \frac{C_{\text{presentedcapacity}}}{C_{\text{initialcapacity}}} \times 100\%, \tag{1}$$

where $C_{\text{presentedcapacity}}$ and $C_{\text{initialcapacity}}$ are the discharging capacity of the present time and the initial time, respectively.

## 3. Chi-Squared Statistic

Chi-square is a statistic of quantitative measuring the correlation or independence of two variables, which was introduced by Pearson in 1900. Since the chi-squared statistic was proposed, it has been extensively used in many other fields of science, such as image denoising [30], and signal recognition [31]. The chi-squared statistic could offer a better characterization of dependency for the observed data. The classical chi-squared statistic is defined as [32]:

$$S = \sum_{i=1}^{n} \frac{(x_i - \overline{x_i})^2}{\overline{x_i}}, \tag{2}$$

where $x_i$ is the sampling voltages of the constant current-constant voltage charging phrase for the cycle $i$, and $n$ represents the number of battery charging and discharging cycles; $\overline{x_i}$ is the average voltage of constant current-constant voltage charging phrase for the cycle $i$.

A large amount of observed battery data is always required for accurate SOH estimation, but the measured battery data are often affected by all kinds of noise pollution. It is complicated and imprecise to characterize the battery health based on the massive noisy acquired data. Therefore, it is significantly important to extract the key characteristics from the noisy acquired battery data. The chi-squared test is used to quantify the correlation and independence of the battery data and simplify the calculation through makes hundreds or thousands of sampling points into a chi-squared statistic value in this paper.

## 4. ELM-LSTM Algorithm

### 4.1. LSTM Neural Network

Since the SOH degradation of the battery varies with the cycles of charging and discharging, it could be considered as a time series process. Therefore, the LSTM neural network, a time series analysis approach, is applied to estimate the SOH value based on the historical operation data for the batteries. LSTM neural network is a type of recurrent neural network (RNN), which is skilled in setting sequential data. A simple RNN structure is showed in Figure 1, whose recurrent layer is unfolded into a full network, where $x$ is input, and the hidden state $h$ gives the network memory ability, $t$ represent time step and $w$, $u$ and $v$ are parameters of different layers.

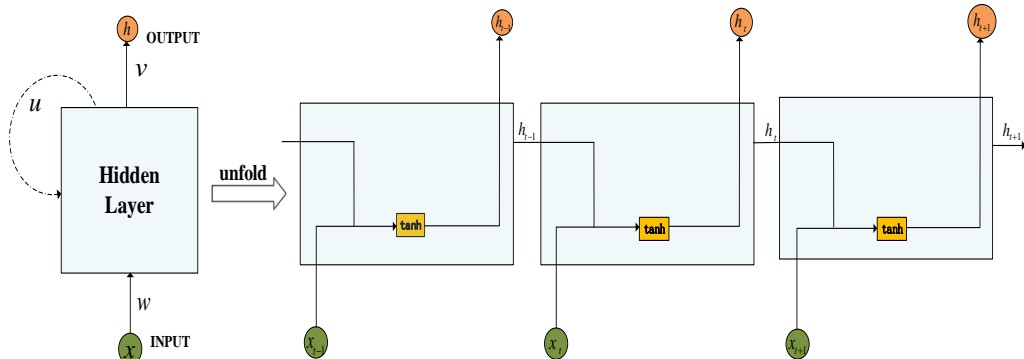

**Figure 1.** The structure of RNN.

It can be observed that the output of the hidden layer with the present information state is transferred to the hidden layer of the next step time as part of the input. This distinctive characteristic preserves the message of the previous step, thereby improving the learning ability for the time series data. However, the gradient of RNN vanishes for the long-term dependence problem, meaning that the input cannot be far transferred as a result.

In terms of the gradient vanishing problem of RNN, the most effective solution so far is the LSTM cell architecture. The key insight in the LSTM design was to incorporate nonlinear, data-dependent controls into the RNN cell, which can be trained to ensure the gradient of the objective function concerning the state signal does not disappear. The LSTM structure is showed in Figure 2. It is obvious that the inner structure of the hidden layer for the LSTM is more complex than that of RNN. The LSTM mainly includes memory cell state, forget gate, input gate, and output gate. The forget gate can discard redundant information; the input gate is able to select key information to be stored in the internal state; and the output gate is used to determine output information. Adding or removing information is carried out selectively by the memory cell state with the help of three gates. Accordingly, LSTM can efficiently store and update key information over a long period of time without gradients vanishing.

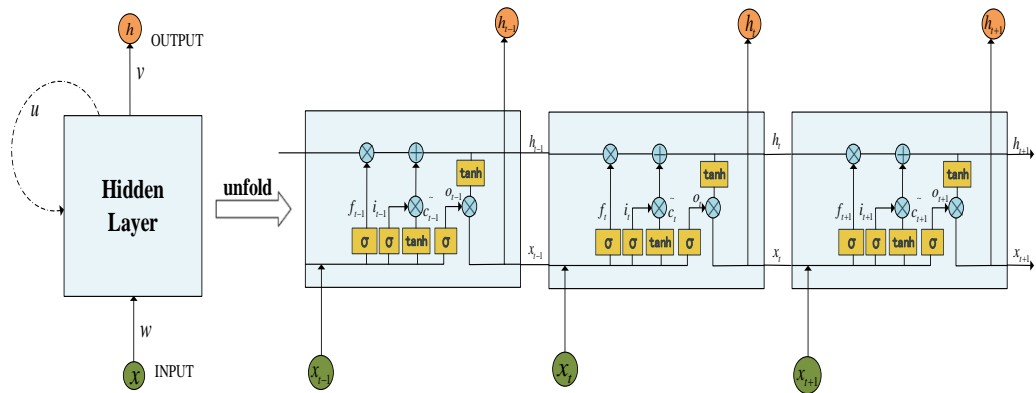

**Figure 2.** The structure of LSTM neural network.

The procedures of the measured data processed by the LSTM neural network are described as follows [32]:

(1) Discard the unneeded message of the previous cell state $c_{t-1}$ in the forget gate $f_t$ by

$$f_t = \sigma(w_f x_t + u_f h_{t-1} + b_f). \tag{3}$$

(2) Update the input information $i_t$ and the candidate cell state $\widetilde{c}_t$ through

$$i_t = \sigma(w_i x_t + u_i h_{t-1} + b_i), \tag{4}$$

$$\widetilde{c}_t = \tanh(w_i x_i + u_i h_{t-1} + b_c). \tag{5}$$

(3) Update the cell state of the present time step $c_t$ according to the candidate memory $\widetilde{c}_t$ and the long-term memory $c_{t-1}$:

$$c_t = f_t \cdot c_{t-1} + i_t \cdot \widetilde{c}_t. \tag{6}$$

(4) Generate the outcome $h_t$ according to the output information $o_t$ and the cell state $c_t$ by

$$o_t = \sigma(w_o x_t + u_o h_{t-1} + b_o), \tag{7}$$

$$h_t = o_t \cdot \tanh(c_t), \tag{8}$$

where the $w$ and $u$ are the input weight and recurrent weight, respectively; The $f$, $i$ and $o$ are the forget gate, input gate and output gate, respectively; $b$ is bias, and $\sigma$ is the sigmoid function which activates the three gate and is described as

$$\sigma(x) = \frac{1}{1 + e^{-x}}. \tag{9}$$

The tanh is the hyperbolic tangent function and comprised by

$$\tanh(x) = \frac{e^x - e^{-x}}{e^x + e^{-x}}. \tag{10}$$

### 4.2. ELM Neural Network

The ELM is a new single-hidden layer feedforward neural network learning method. It has the features of adaptive capability, autonomous learning and optimal computation for a large number of unstructured and imprecise laws. It only needs to set the appropriate number of hidden layer nodes before training, and assign random values to the input weights and hidden layer biases during execution. The whole process is completed in one pass without iterations and produces a unique optimal solution [33]. The structure of ELM is shown in Figure 3.

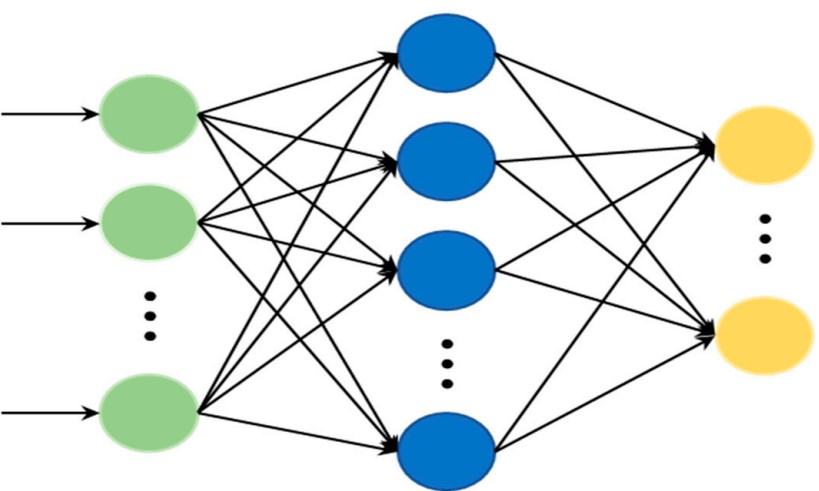

**Figure 3.** The structure of ELM.

The basic flows of ELM algorithm are described as follows [34]:

Step 1: Initialize network parameters randomly. The input weight vector and hidden layer bias are represented as $x_i = [x_{i1}, x_{i2}, \ldots, x_{iN}]^T$ and $b_i$ respectively, where $i$ is the number of neurons in hidden layer. The hidden neurons are assigned as $\widetilde{N}$, and the value of $\widetilde{N}$ can be changed for obtaining reasonable accuracy.

Step 2: Calculate the output matrix of the hidden layer. The mathematical expression is calculated by

$$
\begin{aligned}
H\beta &= T \\
&\Leftrightarrow \sum_{i=1}^{\widetilde{N}} \beta_i f_i(x_i) = \sum_{i=1}^{\widetilde{N}} \beta_i f_i(a_i \cdot x_j + b_j) = t_j, j = 1, \ldots, N
\end{aligned} \tag{11}
$$

where $a_i = [a_{i1}, a_{i2}, \ldots, a_{in}]^T$ represents the weight vector which connects the input nodes and $i$th hidden nodes, $\beta_i = [\beta_{i1}, \beta_{i2}, \ldots, \beta_{in}]^T$ is the output weight which connect the $i$th hidden layer neuron and output layer neuron, $f$ is the activation function which is

determined before training and $H$ is the matrix of the output layer of the neural network determined by randomly allocated input weights and hidden layer biases.

Step 3: Obtain the linear equation $H\beta = T$.

$$\begin{aligned}&\|H(a_1, a_2, \ldots, a_{\widetilde{N}}, b_1, b_2, \ldots, b_{\widetilde{N}})\hat{\beta} - T\|\\&= \min_{\beta}\|H(a_1, a_2, \ldots, a_{\widetilde{N}}, b_1, b_2, \ldots, b_{\widetilde{N}})\hat{\beta} - T\|\end{aligned} \tag{12}$$

Step 4: The least square solution is used to solve the above equation.
The output weight $\beta$ is estimated by

$$\hat{\beta} = H^+ T \tag{13}$$

where $H^+$ is the Moore–Penrose generalized inverse of $H$; the optimal solution $\hat{\beta}$ features the lower training error and optimal generalization performance.

It is clearly observed that ELM needs less computation than other algorithms since it uses a forward pass with series of matrix multiplications, which results in substantial development in training speed.

*4.3. Integrated Approach for the ELM and LSTM Neural Network*

ELM-LSTM algorithm refers to the ELM and LSTM neural network modelled separately and then integrated output at a specific ratio. In this paper, the standard deviation of the error sequence of the integrated modelling training is used as an essential reference for the ratio of the outputs of the two neural networks. The steps to integrating the ELM and LSTM neural network are described as follows:

(1)  Divide the acquired lithium-ion battery aging data into preliminary modelling training set, integrated modelling training set and testing set.
(2)  Initialize the ELM and LSTM neural network parameters, randomly.
(3)  Based on the preliminary modelling training set, the initial lithium-ion battery SOH estimation models are constructed using ELM and LSTM, respectively.
(4)  Calculate the output error series of the preliminary lithium-ion battery SOH estimation model based on the integrated modelling training set, and then obtain the standard deviation of the error series.
(5)  Establish the integrated estimation model of lithium-ion battery SOH, and the output weights of LSTM and ELM are calculated by Equations (14) and (15), respectively.

$$\omega_{lstm} = 1 - \frac{Sde_{lstm}}{Sde_{lstm} + Sde_{elm}} \tag{14}$$

$$\omega_{elm} = 1 - \omega_{lstm} \tag{15}$$

where $Sde_{lstm}$ is the standard deviation of the error series for the LSTM neural network based on the integrated modelling training set, whereas $Sde_{elm}$ represents the standard deviation of the error series for the ELM.

Most of the current literature for battery health state estimation integrates different algorithms using series or by forming feedback through variables [35–37]. However, it is innovative to connect the health state estimation model in parallel with the ELM and LSMT modeling, and then the standard deviation of the modeling error is used to determine the contribution of the final output. The proposed hybrid method incorporates the advantages of both algorithms through this manner, which improves the estimation accuracy consequently. Meanwhile, the aging characteristics of the battery charging data are innovatively extracted using the cardinality statistics. It can be observed from the experimental result that chi-squared statistic could offer a better characterization of dependency for the observed data.

## 5. Experiment Process, Results and Discussions

### 5.1. Experiment Data

Experiments are conducted to demonstrate the effectiveness and universality of the proposed SOH estimation approach. The experiment data adopt lithium-ion batteries data of the public data repository of the NASA Ames Prognostics Center of Excellence and the battery pack aging data. The NASA batteries were employed in working through three different operational profiles: charge, discharge and impedance, with a temperature of 25 °C and 43 °C, which were corresponding to the batteries 5 and 6. Charging was performed at a 1.5 A constant current until the battery terminal voltage reached 4.2 V and then maintaining the 4.2 V constant voltage until the current dropped to 20 mA. Discharging of batteries 5 and 6 were running at a 2 A constant current until the battery voltage felled to 2.7 V and 2.5 V, severally. Impedance measurement was implemented with an electrochemical impedance spectroscopy frequency sweep ranging from 0.1 Hz to 5 kHz. The SOH data of lithium-ion batteries 5 and 6 are showed in Figure 4. It can be noted that the SOH of batteries tends to degrade significantly with charge–discharge cycles. The lithium-ion battery pack, which consists of six high-energy lithium-ion cells of the same specifications connected in series, was tested using the battery pack examination equipment at room temperature in the Anqing Normal University (AQNU) laboratory.

### 5.2. Experiment Procedure

The SOH estimation experiment includes the three cases of the battery 5, battery 6 and battery pack. The specific steps of SOH estimation are showed in Figure 5 and also described as follows:

(1) Calculate the chi-squared statistic of voltage and mean temperature in each charging stage to reflect the capacity loss, and then the SOH data of the batteries are obtained after each discharge stage.

(2) Divide the processed data into preliminary modelling training set, integrated modelling training data and testing set. In this work, the preliminary modelling training set, integrated modelling training data and testing set are divided according to 1:1:2 of the measured data

(3) Based on the preliminary modelling training data, ELM and LSTM neural network, respectively, used for the preliminary modelling.

(4) Establish the integrated SOH estimation model for the lithium-ion battery based on the standard deviation of the error series, which is produced by the preliminary model with the integrated modelling training set as input.

(5) Generate the estimated battery SOH based on the testing data.

### 5.3. Experiment Results and Analysis

The chi-squared and mean temperature are implemented to construct the dataset and yield the processed data, which are divided into preliminary modelling training set, integrated modelling training data and testing set. Based on the preliminary modelling training data, ELM and LSTM neural network, respectively, are used for the preliminary modelling, and then establish the integrated SOH estimation model for the lithium-ion battery based on the standard deviation of the error series, which produces by the preliminary model with the integrated modelling training set as input. The established integrated SOH estimation model is used to estimate SOH for the battery 5, battery 6 and battery pack, respectively. The results are showed in Figure 6. It can be noted that the proposed SOH estimation method proposed in this paper is effective, which accurately estimates the SOH for both batteries with the average error within 1%. The ELM-LSTM ensemble neural network, taking mean temperature and chi-squared as input, can be trained to ensure the gradient does not disappear and the estimated target is precisely obtained, owing to incorporate nonlinear, data-dependent controls into the cell.

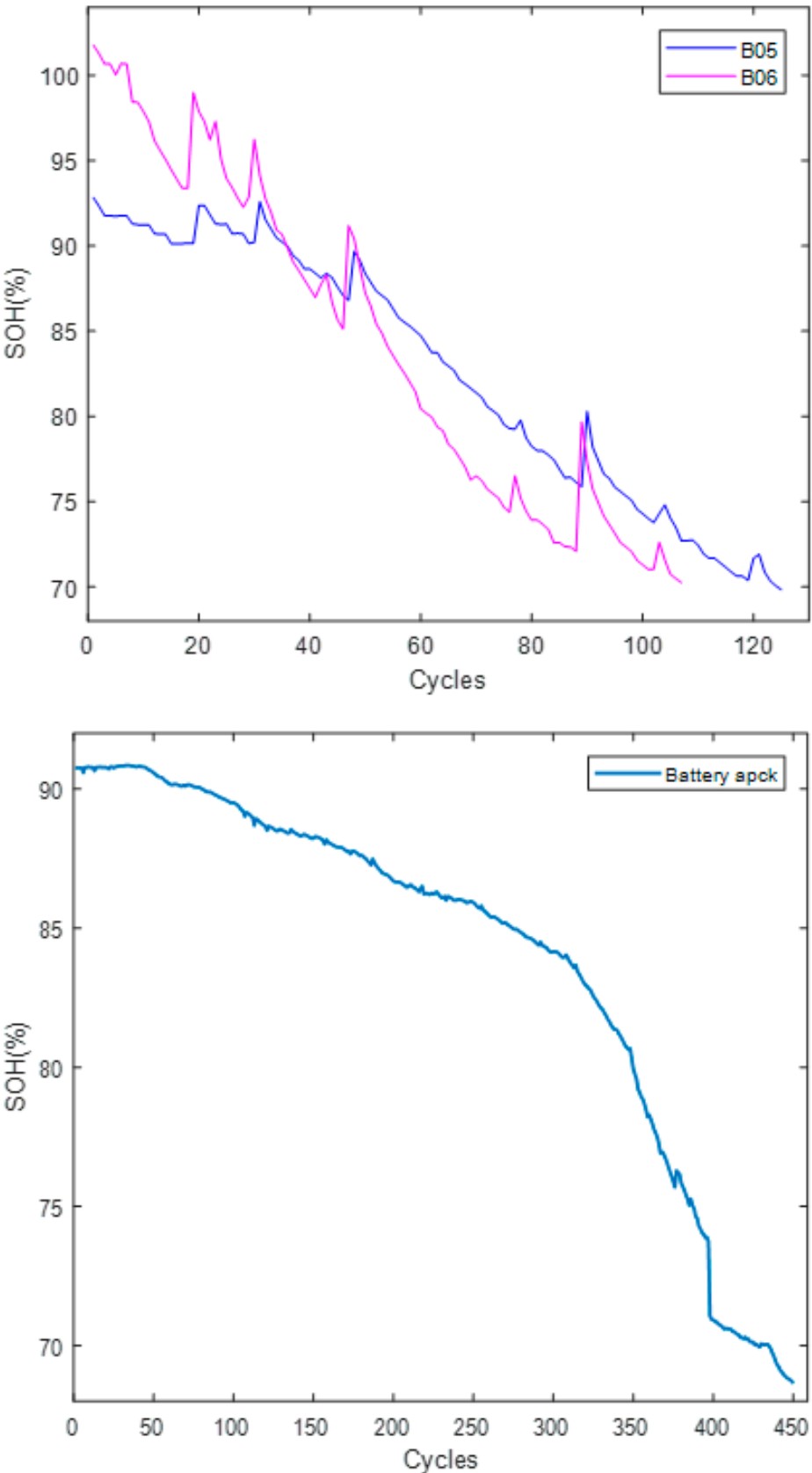

**Figure 4.** Measured SOH data of battery 5, battery 6 and battery pack.

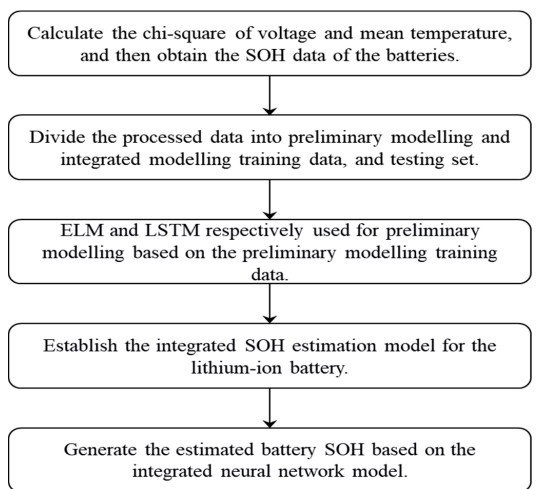

**Figure 5.** Flowchart of estimation steps.

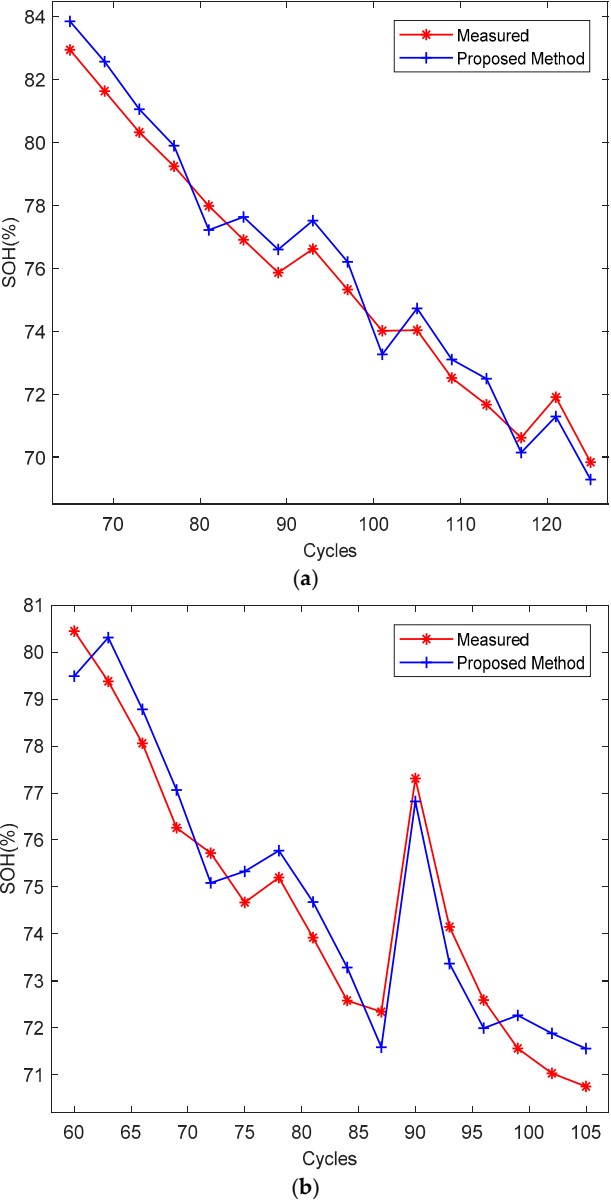

**Figure 6.** *Cont.*

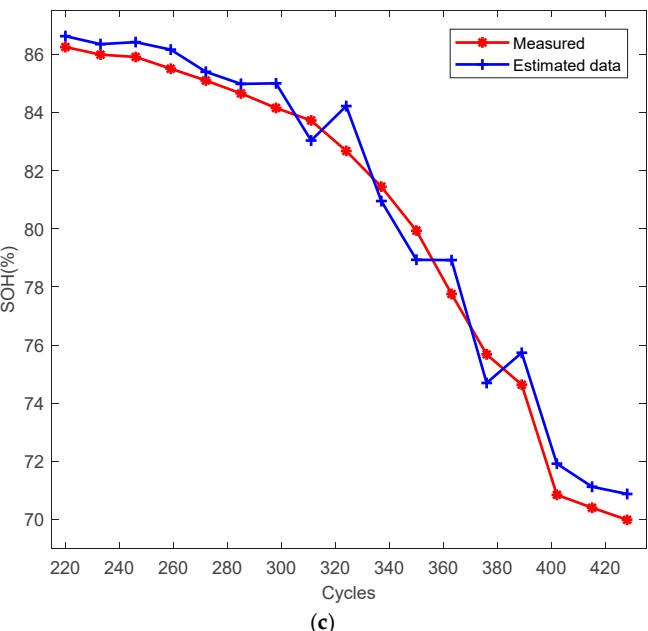

(**c**)

**Figure 6.** Estimation results, (**a**) Battery 5, (**b**) Battery 6, (**c**) Battery pack.

In order to quantify the performance of the presented estimation approach including stability and universality, a comparative experiment was done with the solo ELM neural network and LSTM neural network methods for the cases of battery 5, battery 6 and battery pack, respectively. Meanwhile, to avoid accidental accidents in the experiment, each approach was run more than ten times and estimated results of every time are displayed in Figure 7.

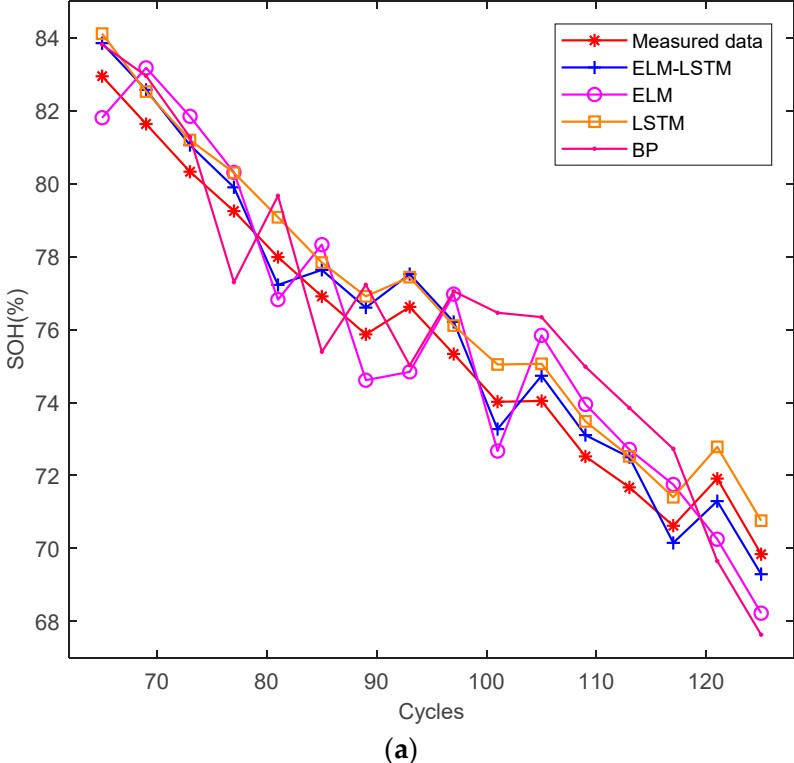

(**a**)

**Figure 7.** *Cont.*

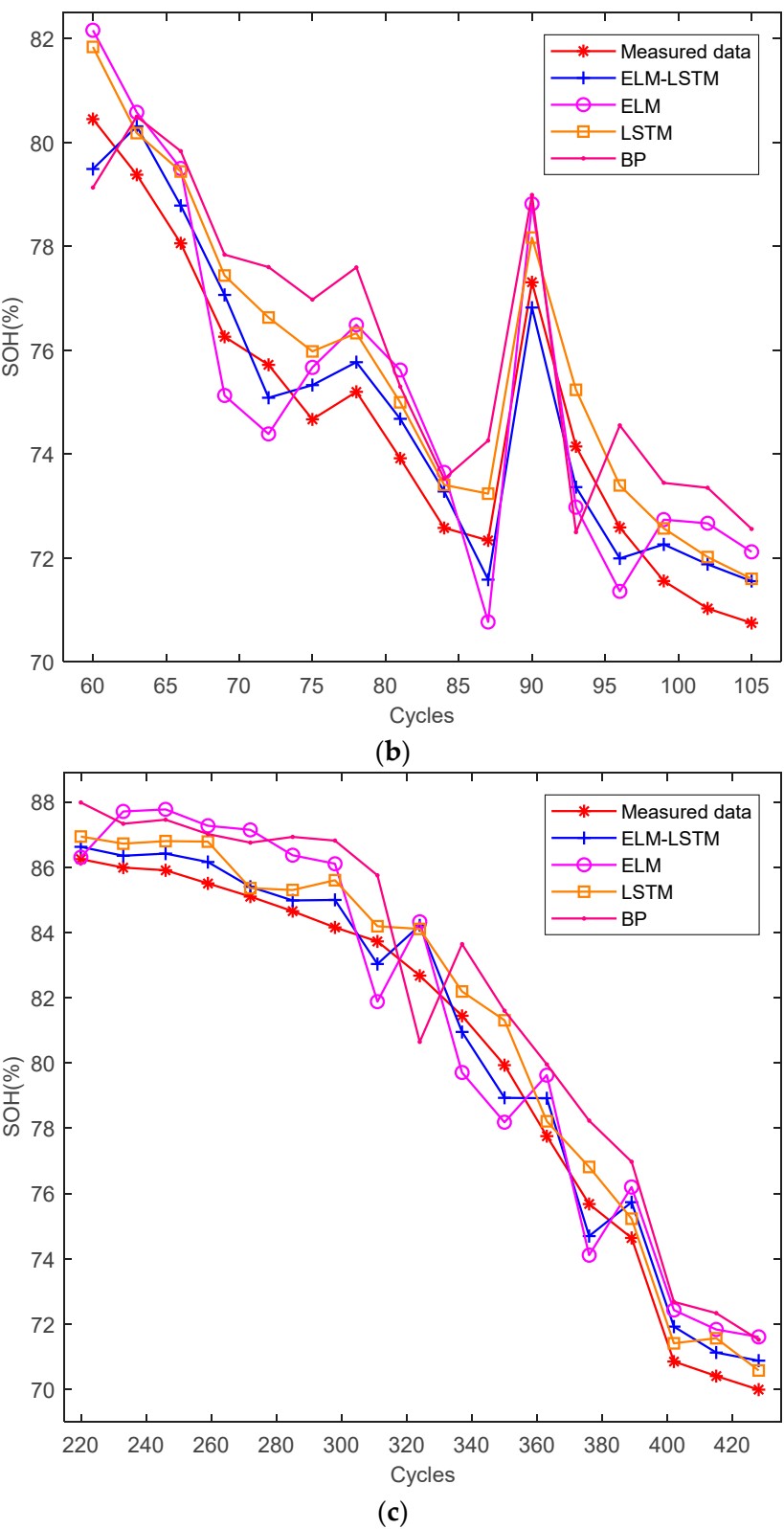

**Figure 7.** Estimation results of the comparative experiment, (**a**) Battery 5, (**b**) Battery 6, (**c**) Battery pack.

Figure 8 shows the errors for the three cases of battery 5, battery 6 and battery pack, respectively. Obviously, it can be observed that the SOH is accurately estimated with the average error (AE) less than 2% in all the two cases, but the proposed method has a better

performance than the other methods. Meanwhile, this can also be verified by the AE and maximal error (ME) in Table 1.

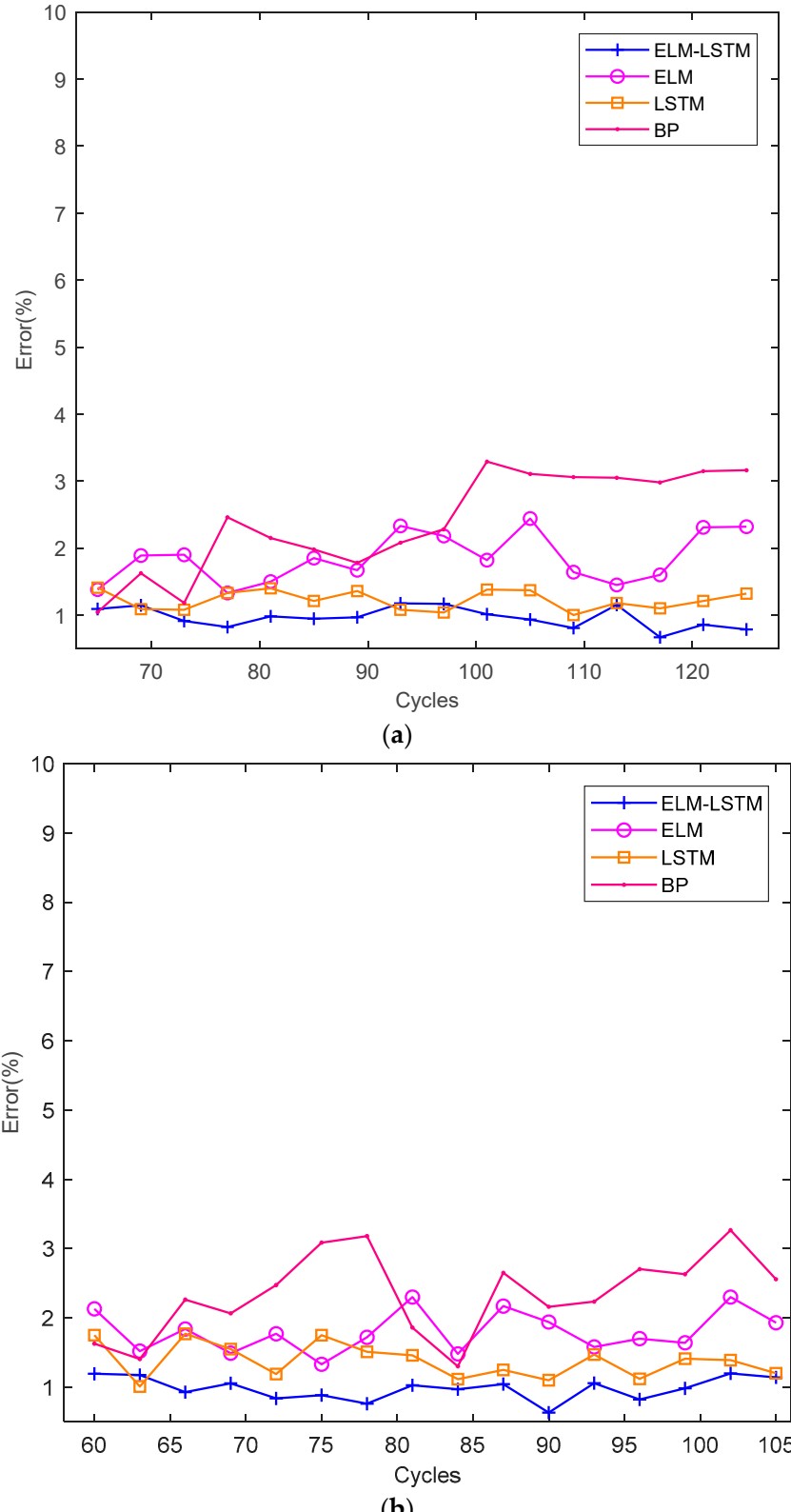

**Figure 8.** *Cont.*

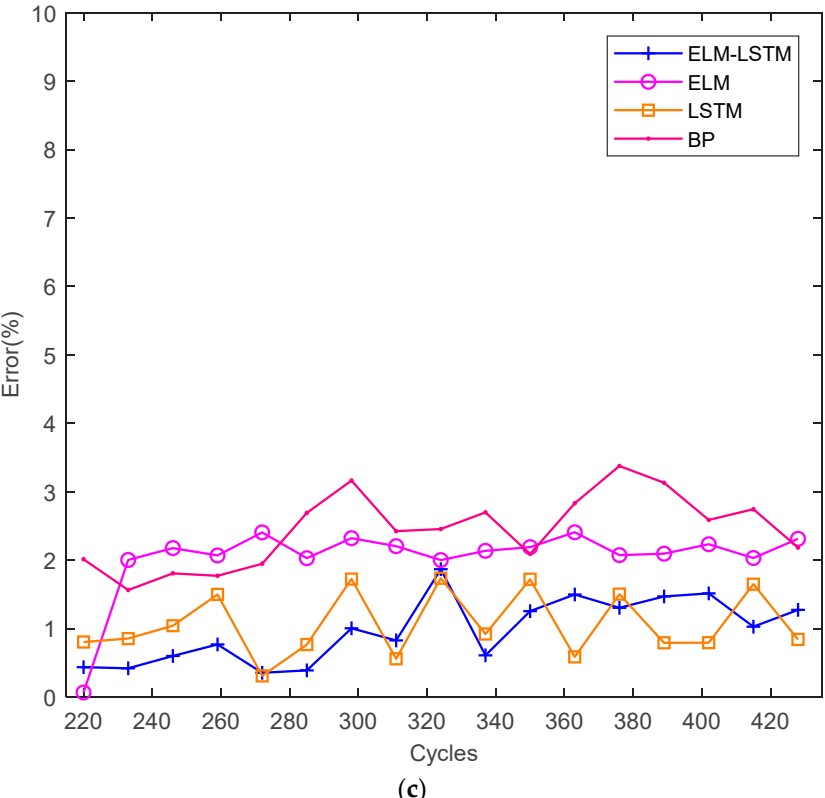

(**c**)

**Figure 8.** Estimation errors of the comparative experiment, (**a**) Battery 5, (**b**) Battery 6, (**c**) Battery pack.

**Table 1.** Estimation errors of the comparative experiment.

| Case | Proposed Method | | ELM Neural Network | | LSTM Neural Network | | BP Neural Network | |
|---|---|---|---|---|---|---|---|---|
| | AE (%) | ME (%) | AE (%) | AE (%) | ME (%) | ME (%) | AE (%) | ME (%) |
| Battery 5 | 0.95 | 1.17 | 1.85 | 1.22 | 1.41 | 2.45 | 2.40 | 3.29 |
| Battery 6 | 0.97 | 1.19 | 1.80 | 1.37 | 1.77 | 2.3 | 2.34 | 3.26 |
| Battery pack | 0.97 | 1.86 | 2.04 | 2.41 | 1.06 | 1.73 | 2.43 | 3.37 |

The proposed integrated neural network has stronger stability according to the estimation results. ELM neural network has high adaptive ability and fast convergence. However, it can be noticed from the estimation results that the processing ability of ELM is obviously not dominant for the time series, for which the AE in the case of cell 5 and cell 6 is 1.85%, 1.80%, and the ME reaches 2.45% and 2.3%, respectively. With the advantage of time series problem processing, the LSTM neural network has a long-time memory function, which can solve the long series training process without the gradient disappearance and gradient explosion problem. AEs of 1.22%, 1.37% along with MEs of 1.41% and 1.77% are obtained for cell 5 and cell 6, separately.

The proposed method is significantly better than BP neural network in accuracy for battery 5, battery 6 and battery pack aging test experiments. In addition, since LSTM neural network incorporates nonlinear, data-dependent control units into the structural framework to ensure that the gradient of the objective function associated with the state signal does not vanish when processing time series problems. At the same time, the connection weights between the implicit and output layers of the ELM do not need to be adjusted iteratively, improving the algorithm accuracy and generalization ability. The proposed method further improves the prediction accuracy by combining the advantages of the two methods, so the proposed method is significantly better than the other methods in terms of estimation results.

Additionally, the conclusion can be obtained from Table 1. The introduced integrated neural network approach combining two neural networks with different characteristics can further improves the accuracy of estimation for the lithium-ion SOH compared to the individual ELM, the separate LSTM neural network and BP neural network. The SOH estimation model for lithium-ion power batteries established by the integrated neural network has a better fitting performance for capacity recession tracking and reduced error. It can be concluded that the ELM-LSTM approach significantly outperforms the individual ELM approach and the separate LSTM approach on the problem of the battery SOH estimation.

## 6. Conclusions

This paper has presented a health estimate approach for lithium-ion batteries based on the synergy of chi-square statistic and the proposed ensemble ELM-LSTM algorithm. The chi-square extracted from battery voltages of each constant current-constant voltage phrase and mean temperature has used as an indicator to characterize the battery SOH loss. Integrated ELM and LSTM neural network has been utilized to capture the underlying correspondence between the SOH, mean temperature and chi-square of battery voltages. NASA battery data and battery pack have been used to demonstrate the procedures of estimation and performance of the proposed approach.

The results have showed that the proposed approach can estimate the battery SOH accurately, and the average estimation error only within 1%. Meanwhile, the comparative experiments have designed to contrast the proposed approach with the solo ELM neural network, the solo LSTM algorithm and BP neural network, and the proposed approach has indicated a better estimation performance in the comparisons.

**Author Contributions:** Methodology, software, validation, formal analysis, writing, J.J.; conceptualization, S.Z.; project administration, C.Z. All authors have read and agreed to the published version of the manuscript.

**Funding:** This work was supported by the National Natural Science Foundation of China under Grant No. 51607004, Natural Science Research Key Project of Education Department of Anhui Province under Grant No. KJ2020A0509, Anhui Provincial Natural Science Foundation under Grant No. 2008085MF197, Collaborative Innovation Project of Anhui Universities Grant under No. GXXT-2019-002.

**Institutional Review Board Statement:** Not applicable.

**Informed Consent Statement:** Informed consent was obtained from all subjects involved in the study.

**Conflicts of Interest:** The authors declare no conflict of interest.

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
