# Peer review of "State-of-Health Estimate for the Lithium-Ion Battery Using Chi-Square and ELM-LSTM"

_wevj, doi:10.3390/wevj12040228_

Round 1

Reviewer 1 Report

This paper has presented a health estimate approach for lithium-ion batteries based on the synergy of chi-square statistic and the proposed ensemble ELM-LSTM algorithm.
(1) There are too few comparative experimental data analysis, which leads to the advantages of the proposed method is not obvious and lacks persuasiveness.
(2) What is the essential difference between the method used and the existing method? Basically, there is no clear explanation. It needs to be explained in principle analysis.
(3) The experimental data adopts NASA, but the data used is not sufficient.

Author Response

Thank you for the reviewers’ comments concerning our manuscript. Those comments are all valuable and very helpful for revising and improving our paper, as well as the important guiding significance to our researches. We have studied comments carefully and have made correction which we hope meet with approval. Revised portion are marked in red in the paper. The responds to the reviewer’s comments are behind the paper.

Reviewer 2 Report

The authors have estimated the state-of health (SOH) with extreme learning machine (ELM) and long short-term memory (LSTM) neural network using the chi-square extracted from battery voltages of each constant current-constant voltage phrase and mean temperature as descriptors. The SOH estimated by this approach had an error of whithin 1%. There are several details of this approach that are unclear.

  1. In Section 4.3 and 5.2, there are duplicate explanations of procedures in the flowchart and the text. Isn't one of them unnecessary?
  2. Was there a difference in the absolute value of the initial capacity between B05 and B06 in Fig. 5?
  3. There is no detailed description of how to divide the data between preliminary modeling training set, integrated modeling training data, and test set. Does the difference in this division affect the prediction accuracy?
  4. The estimation results in Figs. 7 and 8 only show the results after 60 cycles, but what about the prediction accuracy from the first cycle? Is this related to the division of training data and test data?
  5. Can this approach be estimated by extrapolating the data (e.g. after 200 cycles)?

Author Response

(The authors gave the same response as above.)

Round 2

Reviewer 1 Report

The author made revisions and supplements based on the reviewers’ comments, and answered the reviewers’ questions. The quality of the article has been improved. I have no other comments.